# NEAR-OPTIMAL POLICY IDENTIFICATION IN ACTIVE REINFORCEMENT LEARNING

**Xiang Li**[1*]**, Viraj Mehta**[2*]**, Johannes Kirschner**[3*]**, Ian Char**[2]**, Willie Neiswanger**[4]**,
Jeff Schneider**[2]**, Andreas Krause**[1]**, Ilija Bogunovic**[5]

[1]ETH Zurich, [2]Carnegie Mellon Uni., [3]Uni. of Alberta,[4]Stanford Uni., [5]Uni. College London

`xiang.li@outlook.de`, `{virajm,ichar,schneide}@cs.cmu.edu`,
`jkirschn@ualberta.ca`, `neiswanger@cs.stanford.edu`,
`krausea@ethz.ch`, `i.bogunovic@ucl.ac.uk`

## ABSTRACT

Many real-world reinforcement learning tasks require control of complex dynamical systems that involve both costly data acquisition processes and large state spaces. In cases where the transition dynamics can be readily evaluated at specified states (e.g., via a simulator), agents can operate in what is often referred to as planning with a *generative model*. We propose the AE-LSVI algorithm for best-policy identification, a novel variant of the kernelized least-squares value iteration (LSVI) algorithm that combines optimism with pessimism for active exploration (AE). AE-LSVI provably identifies a near-optimal policy *uniformly* over an entire state space and achieves polynomial sample complexity guarantees that are independent of the number of states. When specialized to the recently introduced offline contextual Bayesian optimization setting, our algorithm achieves improved sample complexity bounds. Experimentally, we demonstrate that AE-LSVI outperforms other RL algorithms in a variety of environments when robustness to the initial state is required.

## 1 INTRODUCTION

Reinforcement learning (RL) algorithms are increasingly applied to complex domains such as robotics (Kober et al., 2013), magnetic tokamaks (Seo et al., 2021; Degrave et al., 2022), and molecular search (Simm et al., 2020a;b). A central challenge in such environments is that data acquisition is often a time-consuming and expensive process, or may be infeasible due to safety considerations. A common approach is therefore to train policies offline by interacting with a simulator.

However, even when a simulator is available, such applications require algorithms that are capable of learning and planning in *large state spaces*. Many existing approaches require a large amount of training data to obtain good policies, and efficient active exploration in large state spaces is still an open problem. Moreover, when deploying policies trained on simulators in real-world applications, a crucial requirement is that the policy performs well in *any* state that it might encounter. In particular, at training time, the learning approach has to sufficiently explore the state space. This is of particular importance when at test time, the system's state is partly out of the control of the learning algorithm—e.g., for a self-driving car or robot, which may be influenced by human actions.

In this work, we formally study the setting of *reinforcement learning with a generative model*. Our objective is to learn a near-optimal policy by actively querying the simulator with a state-action pair chosen by the learning algorithm. The simulator then returns a new state that is sampled from the transition model of the (simulated) environment. Inspired by previous works, we make a structural assumption in the kernel setting, which states that the Bellman operator maps any bounded value function to one with a bounded reproducing kernel Hilbert space (RKHS) norm. In particular, this assumption implies that the reward and the optimal $Q$-function can be represented by an RKHS function. We propose a novel approach based on least-squares value iteration (LSVI). The algorithm is designed to actively explore uncertain states based on the uncertainty in the $Q$-estimates, and makes use of optimism for action selection and pessimism for estimating a near-optimal policy.

---

[*]The first three authors contributed equally to this work.

**Contributions** We propose a novel kernelized algorithm for best policy identification in reinforcement learning with a generative model. Our sampling strategy actively explores (i) states for which the best action is the most uncertain and (ii) the corresponding "optimistic" actions. We prove sample complexity guarantees for finding an $\epsilon$-optimal policy *uniformly over any given initial state*. Our bounds scale with the maximum information gain of the corresponding reproducing kernel Hilbert space but *do not* explicitly scale with the number of states or actions. When specialized to the offline contextual Bayesian optimization (BO) setting (Char et al., 2019), we improve upon sample complexity guarantees from prior work. Finally, we include experimental evaluations on several RL and BO benchmark tasks. The former of these includes one of the first empirical evaluations of the model-free optimistic value iteration algorithms with function approximation (Yang et al., 2020).

## 2 PROBLEM STATEMENT

We consider an episodic MDP $\big(\mathcal{S}, \mathcal{A}, H, (\mathbb{P}_h)_{h\in[H]}, (r_h)_{h\in[H]}\big)$ with state space $\mathcal{S}$, action space $\mathcal{A}$, horizon $H \in \mathbb{N}$, Markov transition kernel $(\mathbb{P}_h)_{h\in[H]}$ and deterministic reward functions $(r_h : \mathcal{S} \times \mathcal{A} \to [0,1])_{h\in[H]}$. In particular, for each $h \in [H]$, we let $\mathbb{P}_h(\cdot|s,a)$ denote the probability transition kernel when action $a$ is taken at state $s \in \mathcal{S}$ in step $h \in [H]$. A *policy* consists of $H$ functions $\pi = (\pi_h)_{h\in[H]}$ where for all $h \in [H]$, $\pi_h(\cdot|s)$ is a probability distribution over the action set $\mathcal{A}$. In particular, $\pi_h(a|s)$ is the probability that the agent takes action $a$ in state $s$ at step $h$.

We assume the *generative* (or random) access model, in which the agent interacts with the environment in the following way: Let $T$ denote the number of episodes and $H$ the horizon, i.e., the number of steps in each episode. Then for each $t \in [T]$, $h \in [H]$, the agent chooses $s_h^t \in \mathcal{S}$, $a_h^t \in \mathcal{A}$, obtains the reward $r_h(s_h^t, a_h^t)$ and observes the new state $s_{h,t}' \sim \mathbb{P}_h(\cdot|s_h^t, a_h^t)$.

To measure the performance of an agent, we use the *value* function. For a policy $\pi$, $h \in [H]$, $s \in S$, and $a \in \mathcal{A}$, the value function $V_h^\pi : \mathcal{S} \to \mathbb{R}$ and the $Q$-function $Q_h^\pi : \mathcal{S} \times \mathcal{A} \to [0,H]$ are given by:

$$V_h^\pi(s) = \mathbb{E}_\pi\Big[ \sum_{h'=h}^H r_{h'}(s_{h'}, a_{h'})\Big|s_h = s\Big], \quad Q_h^\pi(s,a) = \mathbb{E}_\pi\Big[ \sum_{h'=h}^H r_{h'}(s_{h'}, a_{h'})\Big|s_h = s, a_h = a\Big], \tag{1}$$

where $\mathbb{E}_\pi$ denotes the expectation with respect to the randomness of the trajectory $\{(s_h, a_h)\}_{h=1}^H$ that is obtained by following the policy $\pi$. We use $\pi^*$ to denote the optimal policy, and we abbreviate $V_h^{\pi^*}, Q_h^{\pi^*}$ as $V_h^*, Q_h^*$, respectively. We also have $V_h^*(s) = \sup_\pi V_h^\pi(s)$ for all $s \in \mathcal{S}$ and $h \in [H]$.

The goal is to find an $\epsilon$-optimal policy while minimizing the number of necessary episodes $T$. More precisely, for a fixed precision $\epsilon > 0$ and horizon $H \in \mathbb{N}$, the goal of the learner is to output a policy $\hat{\pi}_T$ after a suitable number of episodes $T > 0$ such that $\|V_1^* - V_1^{\hat{\pi}_T}\|_{\ell^\infty(\mathcal{S})} \le \epsilon$.

Finally, we also recall the Bellman equation that is associated to some policy $\pi$:

$$V_{H+1}^\pi = 0, \quad Q_h^\pi(s,a) = r_h(s,a) + \mathbb{E}_{s'\sim\mathbb{P}_h(\cdot|s,a)}[V_{h+1}^\pi(s')], \quad V_h^\pi(s) = \mathbb{E}_{a\sim\pi_h(a|s)}[Q_h^\pi(s,a)], \tag{2}$$

and the Bellman optimality equation:

$$V_{H+1}^* = 0, \quad Q_h^*(s,a) = r_h(s,a) + \mathbb{E}_{s'\sim\mathbb{P}_h(\cdot|s,a)}[V_{h+1}^*(s')], \quad V_h^*(s) = \max_{a\in\mathcal{A}} Q_h^*(s,a). \tag{3}$$

It follows that the optimal policy $\pi^*$ is the greedy policy with respect to $\{Q_h^*\}_{h\in[H]}$, a property that is going to be useful later on when defining our active exploration strategy. We use the reproducing kernel Hilbert space (RKHS) function class to represent functions such as the reward functions $\{r_h\}_{h\in[H]}$ and the optimal $Q$-functions $\{Q_h^*\}_{h\in[H]}$ (see the formal statement in Assumption 1). In particular, we consider a space of well-behaved functions defined on $\mathcal{X} = \mathcal{S} \times \mathcal{A}$, where $\mathcal{H}$ denotes an RKHS defined on $\mathcal{X}$ induced by some continuous, positive definite kernel function $k : \mathcal{X} \times \mathcal{X} \to \mathbb{R}$. We also assume that (i) $\mathcal{X} \subset \mathbb{R}^d$ is a compact set, (ii) the kernel function is bounded $k(x, x') \le 1$ for all $x, x' \in \mathcal{X}$, and (iii) every $f \in \mathcal{H}$ has a bounded RKHS norm, i.e., $\|f\|_\mathcal{H} \le B_Q H$ for some fixed positive constant $B_Q > 0$.

## 3 AE-LSVI ALGORITHM

Our algorithm runs in episodes $t \in [T]$ of horizon $H$. As in the kernel least-squares value iteration (Yang et al., 2020), at the beginning of every episode $t$, it solves a sequence of kernel ridge

regression problems based on the data obtained in the previous $t - 1$ episodes to obtain value function estimates $\{\hat{Q}_h^t\}_{h=1}^H$:

$$\hat{Q}_h^t \in \underset{f \in \mathcal{H}}{\arg\min} \Big\{ \sum_{i=1}^{t-1} \big( r_h(s_h^i, a_h^i) + V_{h+1}^t(s_{h,i}') - f(s_h^i, a_h^i) \big)^2 + \lambda \|f\|_{\mathcal{H}}^2 \Big\}, \tag{4}$$

where $\lambda$ is the regularization parameter. Recalling that $x \in \mathcal{X} := \mathcal{S} \times \mathcal{A}$, the solution of the problem in Eq. (4) can be written in closed form as follows:

$$\hat{Q}_h^t(x) := k_h^t(x)^T (K_h^t + \lambda I)^{-1} Y_h^t, \tag{5}$$

where $k_h^t(x) \in \mathbb{R}^{t-1}$, the kernel matrix $K_h^t \in \mathbb{R}^{(t-1) \times (t-1)}$ and observations $Y_h^t \in \mathbb{R}^{t-1}$ are given as follows:

$$k_h^t(x) := [k(x_h^1, x), \dots, k(x_h^{t-1}, x)], \ K_h^t := \big[ k(x_h^i, x_h^{i'}) \big]_{i,i' \in [t-1]}, \ [Y_h^t]_i := r_h(s_h^i, a_h^i) + V_{h+1}^t(s_{h,i}').$$

Next, we can also compute the uncertainty function $\sigma_h^t(\cdot, \cdot)$ in the closed form:

$$\sigma_h^t(s, a) = \frac{1}{\lambda^{1/2}} \big( k(x, x) - k_h^t(x)^T (K_h^t + \lambda I)^{-1} k_h^t(x) \big)^{1/2}. \tag{6}$$

We recall that each reward function is bounded in $[0, 1]$. We use $[\,\cdot\,]_0^{H-h+1}$ to denote the truncation to the interval $[0, H - h + 1]$ and we define the optimistic $\overline{Q}_h^t$ and pessimistic $\underline{Q}_h^t$ value estimates (i.e., upper and lower confidence bound of $Q_h^*$; see Lemma 4.1 and Lemma A.1):

$$\overline{Q}_h^t(\cdot, \cdot) := \big[ \hat{Q}_h^t(\cdot, \cdot) + \beta \sigma_h^t(\cdot, \cdot) \big]_0^{H-h+1}, \quad \overline{V}_h^t(\cdot) := \max_{a \in \mathcal{A}} \overline{Q}_h^t(\cdot, a), \tag{7}$$

$$\hat{Q}_h^t(\cdot) := k_h^t(\cdot)^T (K_h^t + \lambda I)^{-1} \overline{Y}_h^t, \quad [\overline{Y}_h^t]_i := r_h(s_h^i, a_h^i) + \overline{V}_{h+1}^t(s_{h,i}'). \tag{8}$$

Similarly, we have

$$\underline{Q}_h^t(\cdot, \cdot) := \big[ \check{Q}_h^t(\cdot, \cdot) - \beta \sigma_h^t(\cdot, \cdot) \big]_0^{H-h+1}, \quad \underline{V}_h^t(\cdot) := \max_{a \in \mathcal{A}} \underline{Q}_h^t(\cdot, a), \tag{9}$$

$$\check{Q}_h^t(\cdot) := k_h^t(\cdot)^T (K_h^t + \lambda I)^{-1} \underline{Y}_h^t, \quad [\underline{Y}_h^t]_i := r_h(s_h^i, a_h^i) + \underline{V}_{h+1}^t(s_{h,i}'). \tag{10}$$

Our proposed algorithm AE-LSVI is presented in Algorithm 1. At each $h$, the algorithm uses optimistic and pessimistic value estimates from Eqs. (7) and (9) (computed based on the data collected in previous episodes), and selects $s_h^t$ and $a_h^t$ as:

$$s_h^t \in \underset{s \in S}{\arg\max} \Big[ \max_{a \in A} \overline{Q}_h^t(s, a) - \max_{a \in A} \underline{Q}_h^t(s, a) \Big], \tag{11}$$

$$a_h^t \in \underset{a \in \mathcal{A}}{\arg\max} \ \overline{Q}_h^t(s_h^t, a). \tag{12}$$

The main intuition behind the proposed sampling rules is as follows. Since the optimal policy $\pi^*$ is the greedy policy with respect to $\{Q_h^*\}_{h \in [H]}$, we do not need to learn $\{Q_h^*\}_{h \in [H]}$ everywhere on $\mathcal{S} \times \mathcal{A}$. Hence, it is sufficient to focus on discovering the best actions for each state. Our active exploration strategy is explicitly designed to focus on (i) states for which the best action is the most uncertain (Eq. (11)) and (ii) corresponding best "optimistic" actions (Eq. (12)).

We use $\hat{\pi}_T$ to denote the final reported policy returned by AE-LSVI (see Algorithm 1). There are various reasonable greedy-based choices for $\hat{\pi}_T$. The simplest one is to return $\hat{\pi}_{T,h}(\cdot) = \arg\max_{a \in \mathcal{A}} \hat{Q}_h^T(\cdot, a)$, but in our theory and experiments, we focus on equating $\hat{\pi}_T$ with the policy with the highest lower confidence estimate $\underline{Q}_h^t(s, a)$. Our sampling strategy combined with the proposed policy reporting rule allows for discovering an $\epsilon$-optimal policy uniformly over any given initial state as we formally show in the next section.

## 4 THEORETICAL RESULTS

We use the structural assumption for the kernel setting from Yang et al. (2020) which states that the Bellman operator maps any bounded value function to a function with a bounded RKHS norm.

---

**Algorithm 1** AE-LSVI (Active Exploration with Least-Squares Value Iteration)

---

**Require:** kernel function $k(\cdot, \cdot)$, exploration parameter $\beta > 0$, regularizer $\lambda \geq 1$

1: **for** $t = 1, \ldots, T$ **do**
2:      **for** $h \in \{1, \ldots, H\}$ **do**
3:          Set $\overline{Q}_{H+1}^t, Q_{H+1}^t$ as the zero functions
4:          **for** $h = H, \ldots, 1$ **do**
5:              Obtain $\overline{Q}_h^t$ and $Q_h^t$ from Eq. (7) and Eq. (9)
6:          **end for**
7:          Choose $s_h^t \in \arg\max_{s \in S} \left[ \max_{a \in A} \overline{Q}_h^t(s, a) - \max_{a \in A} Q_h^t(s, a) \right]$
8:          Choose $a_h^t \in \arg\max_{a \in \mathcal{A}} \overline{Q}_h^t(s_h^t, a)$
9:          Observe the reward $r_h(s_h^t, a_h^t)$ and the next state $s_{h,t}' \sim \mathbb{P}_h\left( \cdot \,|\, s_h^t, a_h^t \right)$
10:      **end for**
11: **end for**
12: Output the policy estimate $\hat{\pi}_T$ such that $\hat{\pi}_{T,h}(\cdot) = \arg\max_{a \in \mathcal{A}} \max_{t \in [T]} Q_h^t(s, a)$

---

**Assumption 1.** *Let $B_Q > 0$ be a fixed positive constant. Let $k : (\mathcal{S} \times \mathcal{A})^2 \to \mathbb{R}$ be a continuous kernel function on a compact set $\mathcal{S} \times \mathcal{A} \subset \mathbb{R}^d$ such that $\sup_{x, x' \in \mathcal{S} \times \mathcal{A}} k(x, x') \leq 1$. We assume that $\|T_h^* Q\|_{\mathcal{H}} \leq B_Q H$ for all functions $Q : \mathcal{S} \times \mathcal{A} \to [0, H]$ and all $h \in [H]$, where $T_h^*$ denotes the Bellman optimality operator, i.e.,*

$$T_h^* Q(s, a) = r_h(s, a) + \mathbb{E}_{s' \sim \mathbb{P}_h(\cdot|s,a)}\left[ \max_{a' \in \mathcal{A}} Q(s', a') \right]. \tag{13}$$

Assumption 1 implies that for every $h \in [H]$, both $r_h(\cdot, \cdot)$ and $Q_h^*(\cdot, \cdot)$ are elements of the set $\{f \in \mathcal{H} : \|f\|_{\mathcal{H}} \leq B_Q H\}$. Conversely, a sufficient condition for Assumption 1 to be satisfied with $B_Q = 2$ is that $\{r_h(\cdot, \cdot), \mathbb{P}_h(s'|\cdot, \cdot)\} \subseteq \{f \in \mathcal{H} : \|f\|_{\mathcal{H}} \leq 1\}$ for all $h \in [H]$ and $s' \in \mathcal{S}$ (Yang et al., 2020). Moreover, only assuming $Q_h^* \in \mathcal{H}, \|Q_h^*\| \leq B_Q H$ for all $h \in [H]$ is not enough in order to obtain sample size guarantees which are polynomial in $H$ and $d$ (Du et al., 2020).

The main quantity that characterizes the complexity of the RKHS function class in the kernelized setting is the maximum information gain (Srinivas et al., 2010)

$$\Gamma_k(T, \lambda) := \sup_{D \subseteq \mathcal{S} \times \mathcal{A}, |D| \leq T} \tfrac{1}{2} \ln |I + \lambda^{-1} K_{D,D}|, \tag{14}$$

where $K_{D,D}$ denotes the Gram matrix, $|\cdot|$ denotes the determinant, $\lambda > 0$ is a regularization parameter, and the index $k$ indicates the kernel. This quantity is known to be sublinear in $T$ for most of the popularly used kernels (Srinivas et al., 2010).

Further, we define the set of possible optimistic and pessimistic value functions

$$\mathcal{Q}(T, h, b) = \Big\{ Q(\cdot, \cdot) = \big[ \hat{Q}(\cdot, \cdot) \pm \beta \sigma_D(\cdot, \cdot) \big]_0^{H-h+1} :$$
$$\hat{Q} \in \mathcal{H}, \|\hat{Q}\|_{\mathcal{H}} \leq 2H \sqrt{\Gamma_k(T, \lambda)}, \beta \in [0, b], D \subseteq \mathcal{S} \times \mathcal{A}, |D| \leq T \Big\}, \tag{15}$$

where $b > 0$ and $\sigma_D(\cdot, \cdot)$ is of the form Eq. (6) computed with a data set $D \subseteq \mathcal{S} \times \mathcal{A}$, and denote its $\ell^\infty$-covering number as $N_\infty(\epsilon, T, h, b)$. [1] Our sample complexity bounds depend on $b_T > 0$ defined as the smallest number that satisfies the following inequality:

$$8\Gamma_k\big(T, \tfrac{T+1}{T}\big) + 8 \log N_\infty(H/T, T, h, b_T) + 16 \log(2TH) + 22 + 2B_Q^2\big(\tfrac{T+1}{T}\big) \leq (b_T/H)^2 \tag{16}$$

For many kernel functions, $b_T$ has a sublinear dependence on $T$. For instance, $b_T = \mathcal{O}(\gamma H \sqrt{\log(\gamma TH)})$ for bounded and continuously differentiable kernels with $\gamma$-finite spectrum and $b_T = \mathcal{O}(H\sqrt{TH} \log(T)^{1/\gamma})$ for bounded and continuously differentiable kernels with $\gamma$-exponential decay. See (Yang et al., 2020, Corollary 4.4) for more details.

---

[1] The results on $b_T$ hold despite Yang et al. (2020, Lemma D.1) being stated only in the case of the smaller class obtained from only *adding* $+\beta\sigma_D(\cdot, \cdot)$ in the definition of $Q(\cdot, \cdot)$ in $\mathcal{Q}(T, h, b)$ in Eq. (15).

We recall that the Bellman equation implies that $\overline{Q}_{h+1}^t(\cdot)$, $\underline{Q}_h^t(\cdot)$ are upper and lower confidence bounds for $Q_h^*$ for all $h \in [H]$, respectively (see Lemma A.1), while the target functions of kernel ridge regressions are $T_h^* \overline{Q}_{h+1}^t(\cdot)$ and $T_h^* \underline{Q}_{h+1}^t(\cdot)$. As a technical tool, we use the following concentration result that follows from (Yang et al., 2020, Lemma 5.2).

**Lemma 4.1.** *Consider the setup of Assumption 1, and $\overline{Q}_{h+1}^t(\cdot)$, $\underline{Q}_h^t(\cdot)$ $\sigma_h^t(\cdot)$ from Eqs. (6), (7) and (9) computed with $\lambda = 1 + 1/T$ and $\beta = b_T$ from Eq. (16). Then with probability at least $1 - (2T^2 H^2)^{-1}$, the following holds for all $t \in [T]$, $h \in [H]$ and all $(s, a) \in \mathcal{S} \times \mathcal{A}$:*

$$0 \leq \overline{Q}_h^t(s, a) - T_h^* \overline{Q}_{h+1}^t(s, a) \leq 2\beta \sigma_h^t(s, a), \tag{17}$$

$$0 \leq T_h^* \underline{Q}_{h+1}^t(s, a) - \underline{Q}_h^t(s, a) \leq 2\beta \sigma_h^t(s, a). \tag{18}$$

With the previous confidence lemma in place, we state our main theorem that characterizes the sample complexity of AE-LSVI . The proof is given in Appendix A.2.

**Theorem 4.2.** *Consider the setting of Lemma 4.1 and let $H \in \mathbb{N}$ be a fixed horizon. When running Algorithm 1 for $T$ episodes, then with probability at least $1 - (2T^2 H^2)^{-1}$, the best-policy estimate $\hat{\pi}_T$ (Algorithm 1, Line 12) satisfies:*

$$\|V_1^* - V_1^{\hat{\pi}_T}\|_{\ell^\infty(\mathcal{S})} \leq 2\sqrt{3}\beta H(H+1)\sqrt{\frac{\Gamma_k(T,\lambda)}{T}}. \tag{19}$$

*In other words, for a given fixed precision $\epsilon > 0$, after $T = O\left(\frac{\beta^2 H^4 \Gamma_k(T,\lambda)}{\epsilon^2}\right)$ episodes (or $O\left(\frac{\beta^2 H^5 \Gamma_k(T,\lambda)}{\epsilon^2}\right)$ samples) $\|V_1^* - V_1^{\hat{\pi}_T}\|_{\ell^\infty(\mathcal{S})} \leq \epsilon$ holds with probability at least $1 - (2T^2 H^2)^{-1}$.*

The obtained result is general since it holds for any kernel function that satisfies Assumption 1. To obtain concrete kernel-dependent regret bounds it remains to specify the kernel and the bounds for the corresponding maximum information gain in Eq. (16). These are summarized in Yang et al. (2020) for the most widely used kernels (see Assumption 4.3 and its discussion).

In the special case of linear kernels with the feature dimension $d$, our sample complexity guarantee reduces to $\tilde{O}(\frac{d^3 H^7}{\epsilon^2})$. Better bounds (in terms of $d$) for this special case are known $\tilde{O}(\frac{d^2 H^7}{\epsilon^2})$, see, e.g., Agarwal et al. (2019, Theorem 3.3). These bounds are obtained by the LSVI algorithm with D-optimal design. Unlike this algorithm, AE-LSVI uses optimism for active exploration and such a performance gap is present even in the simpler linear bandit setting where optimistic algorithms are known to attain worse sample complexity guarantees (Lattimore & Szepesvári, 2020, Chapter 22). The special case also includes the linear MDP setting, which assumes linear reward functions and linear transition kernels. For linear MDPs it is possible to find a policy $\pi$ satisfying $V_1(s_1) - V_1^\pi(s_1) \leq \epsilon$ using $\tilde{O}(d^2 H^3/\epsilon^2)$ samples (Hu et al., 2022); in our setting of Assumption 1, such a policy $\pi$ can be found using $O(H^5 \beta^2 \Gamma_k(T,\lambda)/\epsilon^2)$ samples (Yang et al., 2020). Both results hold with at least a constant probability. However, they require that the initial state $s_1$ is fixed for all episodes. In contrast, the result of Theorem 4.2 holds uniformly over the entire state space.

## 5 APPLICATION TO OFFLINE CONTEXTUAL BAYESIAN OPTIMIZATION

In this section, we specialize Algorithm 1 to the offline contextual Bayesian optimization setting (Char et al., 2019). We show that in this setting the proposed active exploration scheme leads to new sample complexity bounds that hold *uniformly* over the context space.

The offline contextual Bayesian optimization setting is similar to the one considered in Section 2 when $H = 1$. In particular, instead of having $H$ different functions to learn, we have a single unknown objective $Q^* : \mathcal{S} \times \mathcal{A} \to \mathbb{R}$ that we learn about (from noisy point evaluations). Here, we refer to $\mathcal{S}$ as the context space, and assume that both $\mathcal{S}$ and $\mathcal{A}$ are compact sets. As before, we use a shorthand notation $\mathcal{X} = \mathcal{S} \times \mathcal{A}$. In each round $t \in [T]$, the learner chooses a context-action pair $(s^t, a^t) \in \mathcal{S} \times \mathcal{A}$ and observes $y_t = Q^*(s^t, a^t) + \eta_t$ (with independent sub-Gaussian noise). To choose $(s^t, a^t)$ at each round $t$, we make use of the same active exploration strategy from Eqs. (11) and (12). Our complete algorithm for the offline BO setting can be found in Appendix A.3 (see Algorithm 2).

We define $\hat{Q}^t : \mathcal{S} \times \mathcal{A} \to \mathbb{R}$ (and $\sigma^t : \mathcal{S} \times \mathcal{A} \to \mathbb{R}$) similarly as $\hat{Q}_h^t$ (resp. $\sigma_h^t$) from Eq. (5) (resp. Eq. (6)) but with the modification of ignoring the index $h$ and defining $Y_t := (y_i)_{i=1}^{t-1} \in \mathbb{R}^{t-1}$. We

further define the upper and lower confidence bounds for $Q^*$ as:

$$\overline{Q}^t(\cdot, \cdot) = \hat{Q}^t(\cdot, \cdot) + \beta_t \sigma^t(\cdot, \cdot), \quad \underline{Q}^t(\cdot, \cdot) = \hat{Q}^t(\cdot, \cdot) - \beta_t \sigma^t(\cdot, \cdot). \tag{20}$$

When $Q^* \in \mathcal{H}$ and $\|Q^*\|_{\mathcal{H}} \leq B$ correspond to some known kernel (such that $k(x, x') \leq 1$ for all $x, x' \in \mathcal{X}$), then $(\beta_t)_{t \in [T]}$ is a non-decreasing sequence of parameters that can be chosen according to Abbasi-Yadkori (2012, Theorem 3.11) to yield valid confidence bounds. Similarly, in case of $Q^* \sim \mathrm{GP}_{\mathcal{X}}(0, k)$ (Bayesian setting), we can utilize Gaussian Process confidence bounds (Srinivas et al., 2010) and use the corresponding $(\beta_t)_{t \in [T]}$ sequence. In what follows, we assume that $(\beta_t(\delta))_{t \in [T]}$ is a non-decreasing sequence such that with probability at least $1 - \delta$,

$$\underline{Q}^t(s, a) \leq Q^*(s, a) \leq \overline{Q}^t(s, a) \tag{21}$$

holds for all $t \in [T]$ and $(s, a) \in \mathcal{S} \times \mathcal{A}$.

**Corollary 5.1.** *Assume* $(\beta_t(\delta))_{t \in [T]}$ *is set to satisfy Eq.* (21). *Fix* $\epsilon \in (0, 1)$ *and run Algorithm 2 for*

$$T \geq \frac{12\beta_T^2 \Gamma_k(T, \lambda)}{\epsilon^2} \tag{22}$$

*rounds. Then, for every* $s \in \mathcal{S}$, *the reported policy* $\hat{\pi}_T(\cdot)$ *computed as in Line 6 (Algorithm 2) satisfies* $Q^*(s, \hat{\pi}_T(s)) \geq \max_{a \in \mathcal{A}} Q^*(s, a) - \epsilon$ *with probability at least* $1 - \delta$.

We briefly compare the result obtained in Corollary 5.1 with related results from the literature. In the Bayesian setting, Char et al. (2019, Theorem 1) obtain a sample complexity that scales as $\mathbb{E}[T] = O(|\mathcal{S}|^3 |\mathcal{A}| \Gamma_k(T, \lambda)/\epsilon^2)$ in expectation for a given context distribution. In comparison, our result obtained in Eq. (22) holds in $\ell^\infty$-norm over the context space (i.e., implies bounds for *any* context distribution). When specialized to the finite set $\mathcal{X} = \mathcal{S} \times \mathcal{A}$ and when $f \sim \mathrm{GP}_{\mathcal{X}}(0, k)$, the result of Corollary 5.1 holds with $\beta_T = O(\log(|\mathcal{X}|T^2))$ (Srinivas et al., 2010), which then results in $T = O\left(\frac{\log^2(|\mathcal{X}|T^2)\Gamma_k(T, \lambda)}{\epsilon^2}\right)$ leading to a significant improvement for large discrete context spaces. In the setting of distributionally robust Bayesian optimization (DRBO), Kirschner et al. (2020) obtain a result with the same dependency as ours. However, their bound holds only for a *fixed* contextual distribution and degenerates as a function of the distance between the training and test distributions.

## 6 RELATED WORK

Reinforcement learning with function approximation dates back to at least (Bellman et al., 1963; Daniel, 1976; Schweitzer & Seidmann, 1985). A majority of work is in the *online* setting where the learning agent interacts with the environment while (typically) minimizing regret. Upper confidence bound algorithms, originally developed in the bandit setting (Lattimore & Szepesvári, 2020) (also, frequently used in the related setting of best-arm identification, e.g., (Gabillon et al., 2011; Kalyanakrishnan et al., 2012; Soare et al., 2014)) , have been successfully applied to tabular Markov decision processes (MDPs) (Auer & Ortner, 2006; Auer et al., 2008), and extended to RL with function approximation. Jin et al. (2020) propose the LSVI-UCB algorithm in the linear MDP setting that achieves a near-optimal regret bound. Yang et al. (2020); Domingues et al. (2021) extend this work to the non-linear function approximation setting. These works are closely related to ours in that we make use of LSVI and confidence bounds for the $Q$-function in the kernelized setting. Unlike previous works, we consider the generative model setting and derive bounds on the sample complexity that hold uniformly over the initial state. There are many more alternative parametric models that admit sample efficient algorithms (e.g., Ayoub et al., 2020; Zhou et al., 2021; Du et al., 2021; Zanette et al., 2020; Liu & Su, 2022).

In the *generative model* setting, the learner has access to a simulator that for any given state-action pair returns a next-state sample from the transition kernel. This provides additional flexibility to obtain data from states that are otherwise hard to reach in the environment. For the tabular case, matching upper and lower bounds are shown by Azar et al. (2012); Gheshlaghi Azar et al. (2013). In the generative model setting with function approximation, Lattimore et al. (2020) show that policy iteration can be used to compute a near-optimal policy given features such that the $Q$-function of any policy can be approximated by a linear function. Their algorithm uses a D-experimental design to roll out policies from a sufficiently diverse set of states. The POLITEX algorithm (Abbasi-Yadkori et al., 2019; Szepesvári, 2022) can be used in lieu of policy iteration and leads to tighter bounds on

the approximation error. A similar approach based on LSVI is analyzed by Agarwal et al. (2019, Chapter 3).

In practical applications of RL, simpler approaches to exploration are often used or exploration techniques inspired by upper-confidence bound algorithms or Thompson sampling are combined with deep learning function approximation. To list a few, the $\epsilon$-greedy approach (Mnih et al., 2013), upper confidence bounds (UCB) (Chen et al., 2017), Thompson sampling (TS) (Osband et al., 2016a), added Ornstein-Uhlenbeck action noise (Lillicrap et al., 2015), and entropy bonuses (Haarnoja et al., 2018) are all widely applied. More sophisticated methods actively plan to encounter *novel* states (Shyam et al., 2019; Ecoffet et al., 2021). Though these methods serve as reasonable heuristics and are usually computationally efficient, they either lack theoretical guarantees or lead to methods that require large numbers of samples. One recent practical work (Mehta et al., 2022b) gives an *acquisition function* for the generative model setting based on methods from Bayesian experimental design (Neiswanger et al., 2021), and achieves good policies with small numbers of samples; however, this model-based method assumes access to the MDP reward function and is computationally expensive.

An important special case of the MDP setting is the *contextual bandit setting*. When combined with linear function approximation, this recovers the contextual linear bandit setting (Abbasi-Yadkori et al., 2011), and contextual Bayesian optimization when using kernel features (Srinivas et al., 2010; Krause & Ong, 2011). Various works consider the case where the learner has control over the choice of context during training time. Char et al. (2019) propose a variant based on Thompson sampling. Pearce & Branke (2018); Pearce et al. (2020) also propose variants that leverage ideas from the knowledge gradient (Frazier et al., 2009). The latter works lack theoretical guarantees, while our result (from Section 5) improves upon the sample complexity guarantee of Char et al. (2019). The approach by Kirschner et al. (2020) for the distributionally robust setting can be specialized to our setting, in which case they recover similar bounds but only for a fixed context distribution.

## 7 EXPERIMENTS

### 7.1 REINFORCEMENT LEARNING EXPERIMENTS

In the previous sections we presented the AE-LSVI algorithm, which provably identifies a near-optimal policy in polynomial time given access to a generative model of the MDP dynamics. Here, we test the AE-LSVI algorithm empirically, and additionally provide one of the first empirical evaluation of the LSVI-UCB method from Yang et al. (2020) on standard benchmarks. We evaluate AE-LSVI and LSVI-UCB on four MDPs from the literature as well as four synthetic contextual BO problems from Char et al. (2019). We discuss details of our implementation in Appendix B.1.

Each environment has a discrete action space. For continuous environments, we discretize the action space into 10 bins per dimension but model the value function in the original continuous state and action space. All methods besides DDQN are initialized by executing a random policy for two episodes. In between exploration episodes, the pessimistic policy $\hat{\pi}_T$ is evaluated by executing it for 10 episodes in the environment.

**Initial State Distribution** To evaluate the policies found by each method, we must initialize the policy at initial states drawn from some distribution $p_0$ at test time. As AE-LSVI does not explicitly consider the initial state distribution, for each environment we choose both a standard $p_0$ from the literature as well as a an alternate distribution $p_0'$ that is translated in the state space, i.e., $p_0'(s) = p_0(s - \Delta_s)$ for some $\Delta_s$. The alternate distribution allows us evaluate the best policy estimate in an area of state space that is not explicitly given to agents. We evaluate each policy using initial states sampled from $p_0'$ as a proxy for understanding how well the optimal policy has been identified in regions of the state space beyond where it was initialized. We give a complete description of the various $p_0'$ for each environment in Appendix B.2. In Table 1, we present results for each method and environment when initialized on $p_0$, which is the typical setup for training and evaluating RL algorithms in the literature. In Table 2 we present results for each method evaluated for the initial state distribution $p_0'$.

**Comparison Methods** Besides AE-LSVI and LSVI-UCB , we compare against several ablations and methods taken from the literature. As a naive baseline for performance in active exploration, we randomly sample state-action pairs from the MDP, evaluate the next states and rewards, and fit $Q$-functions to that data as in the other methods, executing the policy given by the $Q$-function mean (**Random**). We also perform uncertainty sampling (**US**) on the $Q$-function, choosing state-action pairs at each step that maximize $\sigma_h^t(\cdot, \cdot)$ as in Eq. (6). Additionally, we compare

| Environment | AE-LSVI | Random | US | LSVI-UCB | DDQN | BDQN | Greedy |
|---|---|---|---|---|---|---|---|
| Cartpole | $15.2 \pm 0.5$ | $13.6 \pm 0.5$ | $13.6 \pm 0.6$ | $17.1 \pm 0.7$ | $\mathbf{19.3 \pm 0.7}$ | $\mathbf{19.0 \pm 0.8}$ | $17.2 \pm 0.4$ |
| Navigation | $6.0 \pm 1.7$ | $6.7 \pm 1.4$ | $8.9 \pm 0.7$ | $\mathbf{12.9 \pm 0.2}$ | $7.3 \pm 1.5$ | $7.2 \pm 0.9$ | $10.9 \pm 1.5$ |
| $\beta$ Tracking | $12.7 \pm 0.3$ | $11.6 \pm 0.4$ | $11.7 \pm 0.2$ | $\mathbf{13.8 \pm 0.1}$ | $13.4 \pm 0.2$ | $\mathbf{13.9 \pm 0.1}$ | $12.9 \pm 0.3$ |
| $\beta$ + Rotation | $15.2 \pm 0.6$ | $15.2 \pm 0.6$ | $15.1 \pm 0.4$ | $17.8 \pm 0.1$ | $15.1 \pm 0.4$ | $14.2 \pm 0.8$ | $\mathbf{17.9 \pm 0.1}$ |

Table 1: Average Return $\pm$ standard error of executing the identified best policy on the MDP starting from $p_0$ over 5 seeds after collecting 1000 timesteps of data through the use of a generative model (left of line) or episodes starting from $p_0$ (right of line).

| Environment | AE-LSVI | Random | US | LSVI-UCB | DDQN | BDQN | Greedy |
|---|---|---|---|---|---|---|---|
| Cartpole | $\mathbf{16.8 \pm 0.4}$ | $12.9 \pm 0.4$ | $14.5 \pm 0.3$ | $12.9 \pm 0.3$ $(14.2 \pm 0.6)$ | $15.3 \pm 0.6$ $(13.7 \pm 1.3)$ | $16.1 \pm 0.5$ $(13.0 \pm 1.2)$ | $13.3 \pm 0.5$ $(\mathbf{16.7 \pm 0.2})$ |
| Navigation | $\mathbf{22.3 \pm 0.4}$ | $15.3 \pm 0.8$ | $17.5 \pm 1.3$ | $13.6 \pm 0.6$ $(20.6 \pm 1.1)$ | $17.1 \pm 2.4$ $(18.1 \pm 2.6)$ | $21.4 \pm 1.2$ $(18.4 \pm 2.1)$ | $15.2 \pm 1.6$ $(14.0 \pm 0.8)$ |
| $\beta$ Tracking | $\mathbf{14.0 \pm 0.4}$ | $9.2 \pm 0.9$ | $12.5 \pm 0.1$ | $13.3 \pm 0.3$ $(13.7 \pm 0.2)$ | $\mathbf{13.8 \pm 0.1}$ $(13.7 \pm 0.2)$ | $\mathbf{14.0 \pm 0.1}$ $(13.7 \pm 0.1)$ | $12.5 \pm 0.4$ $(\mathbf{13.8 \pm 0.1})$ |
| $\beta$ + Rotation | $\mathbf{14.3 \pm 0.2}$ | $12.8 \pm 1.4$ | $13.3 \pm 0.5$ | $10.1 \pm 0.4$ $(12.7 \pm 0.3)$ | $12.9 \pm 1.1$ $(13.4 \pm 0.3)$ | $13.7 \pm 0.8$ $(12.7 \pm 1.2)$ | $12.8 \pm 0.7$ $(7.5 \pm 0.2)$ |

Table 2: Average Return $\pm$ standard error of executing the identified best policy on the MDP starting from $p_0'$ over 5 seeds after collecting 1000 timesteps of data through the use of a generative model (left) and online RL methods (right). For online methods, numbers without parentheses refer to training from episodes starting from $p_0$, whereas numbers in parentheses use the uniform distribution on the state space as initial states during training.

against three online RL baselines: the Double DQN algorithm (Van Hasselt et al., 2016) where an epsilon-greedy approach is used for exploration (**DDQN**), the bootstrapped DQN (Osband et al., 2016b) which keeps an ensemble of $Q$-functions and does exploration acting according to a sampled $Q$-function each exploratory rollout (**BDQN**), and a greedy exploration algorithm (**Greedy**) that chooses $\arg\max_a \hat{Q}_h^t(s, a)$ at every step $h$ for a given state $s$ but uses the same value iteration procedure used in the main methods. The experiments are conducted with a default exploration bonus $\beta = 0.5$, however, we also empirically analyze the performance for other $\beta$-values in Appendix B.3.

**Environments**  We evaluate all methods on four environments: a **Cartpole** swing-up problem with dense rewards, a nonlinear **Navigation** problem, and two problems ($\beta$ **Tracking** and $\beta$ **+ Rotation**) in plasma control from Mehta et al. (2022a), in which plasma is driven to a desired target state. We give further information on the environments used in Appendix B.2.

**Results**  As our bound on the value function error uses the $\ell^\infty(\mathcal{S})$-norm, our method provably finds an approximately optimal policy regardless of the initial distribution. The LSVI-UCB method is able to quickly learn a policy for the initial state distribution $p_0$ given at training time, as it is designed to minimize regret on the episodic MDP initialized at $p_0$. This can be seen clearly in Table 1, which shows that after 1000 samples, LSVI-UCB performs the best on nearly every environment. In the online setting when the start state distribution is known, greedy and $\epsilon$-greedy methods like DDQN also perform relatively well. We also see in Table 1 that AE-LSVI does not perform particularly well compared to the online methods given the 1,000-sample budget. This is to be expected, as the online methods naturally collect data that is reachable from $p_0$ and in particular LSVI-UCB is designed to minimize regret on episodes beginning from $p_0$. However, this focus on performing well when starting from $p_0$ comes at the expense of active exploration and identifying the best policy uniformly across the state space.

As shown in Table 2, AE-LSVI outperforms the baselines when evaluated on a *different* initial state distribution $p_0'$, even when the online algorithms are initialized from a uniform initial state distribution $p_0$ during training. This is unsurprising, as AE-LSVI is precisely built for this setting and identifies the best action uniformly across the state space, unlike LSVI-UCB which aims to minimize regret starting from an initial state distribution. We see that uncertainty sampling outperforms a random data selection strategy and is comparable to the online methods. However, as we discuss above (in Section 3), in general it is the uncertainty in the value of the best action at a state and not the uncertainty in the value of a state-action pair that needs to be reduced in order to more efficiently find the best policy. We see that, in general, the online methods perform better on $p_0'$ when they train on episodes uniformly initialized on the state space. This suggests that in these cases, it is helpful to make sure that the evaluation distribution $p_0'$ is supported by the training distribution $p_0$. We also note that (as we describe in Appendix B.2) the maximum possible score on **Navigation** starting from

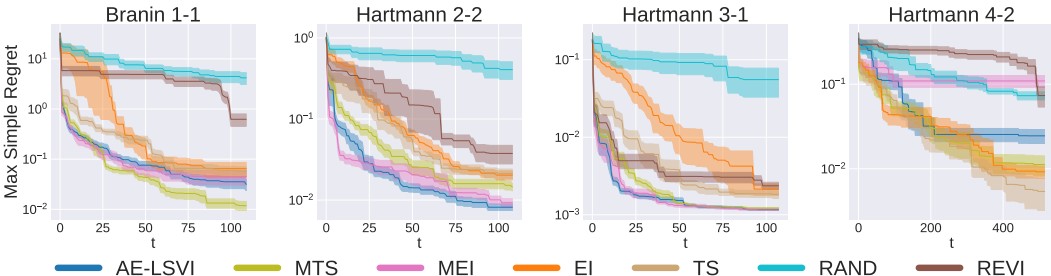

Figure 1: The maximum simple regret seen in any given context for the offline contextual Bayesian optimization experiments. The shaded regions show the standard error over 10 different seeds.

$p'_0$ is higher than that from $p_0$ due to a starting distribution closer to the goal. We believe that these results give empirical support to the theoretical claims of Section 4.

### 7.2 OFFLINE CONTEXTUAL BAYESIAN OPTIMIZATION EXPERIMENTS

We test the performance of AE-LSVI (Algorithm 2 in Appendix A.3) in the offline contextual Bayesian optimization setting. In particular, we test the algorithm on the optimization problems presented in Section 3 of Char et al. (2019), each having a discrete context space but continuous action space. In all experiments, we average over 10 seeds. At the beginning of each experiment, the values corresponding to five actions, chosen uniformly at random, are observed for each context. Every time new data is observed, the hyperparameters of the GP are tuned according to the marginal likelihood. We leverage the Dragonfly library for these experiments (Kandasamy et al., 2020).

**Comparison Methods** For baselines, we compare against the Multi-task Thompson Sampling (**MTS**) method presented by Char et al. (2019), which picks context and action based on the largest improvement over what has been seen according to samples from the posterior. In addition, we compare to the strategy of picking the context with the greatest expected improvement. This method was presented by Swersky et al. (2013), and we refer to it as Multi-task Expected Improvement (**MEI**), following Char et al. (2019). We also compare against the **REVI** algorithm (Pearce & Branke, 2018), which picks contexts and actions that will increase the posterior mean the most across all contexts. Additionally, we show the performance of naive Thompson sampling (**TS**) and expected improvement (**EI**), where contexts are picked in a round robin fashion. Lastly, we show the performance of randomly selecting contexts and actions at each time step (**RAND**).

**Experiment Tasks** To evaluate the method in the case where the objective function is correlated in context space, we take a higher dimensional function and assign some dimensions to context space and the rest to action space. A single GP with a squared exponential kernel is then used to model the objective function. In particular, the Branin-Hoo (Branin, 1972), Hartmann 4, and Hartmann 6 (Picheny et al., 2013) functions are used to create Branin 1-1, Hartmann 2-2, Hartmann 3-1, and Hartmann 4-2, where the first number corresponds to the context dimension and the second to the action dimension. These functions have 10, 9, 8, and 16 equispaced contexts, respectively.

**Results** Figure 1 shows the maximum simple regret seen in any given context as a function of $t$ values observed. As seen from these plots, AE-LSVI often is one of the best performing methods. The only task that AE-LSVI struggles on is Hartmann 4-2. We believe that estimating the amount of improvement to be gained at each context is difficult for this benchmark task. This is supported by the fact none of the more sophisticated methods outperforms the baseline that applies **EI** in a round-robin fashion. It is likely that improved modeling or hyperparameter selection is needed in order for these methods to achieve the highest performance on this task.

## 8 CONCLUSION

We provided a new kernelized least-squares value iteration algorithm for RL in the generative model setting, which aims to learn a near-optimal policy for all initial states by actively exploring states for which the best action is the most uncertain. Our algorithm identifies a near-optimal policy uniformly over the entire state space and attains polynomial sample complexity. Experimentally, we demonstrate that it outperforms other RL algorithms in a variety of environments when robustness to the initial state is required. Perhaps the most immediate direction for future work is to extend the algorithm to the local access model (Yin et al., 2022) in which the simulator can be queried only for states that have been encountered in previous simulation steps.

ACKNOWLEDGMENTS

Johannes Kirschner gratefully acknowledges funding from the SNSF Early Postdoc.Mobility fellowship P2EZP2_199781.

Ian Char is supported by the National Science Foundation Graduate Research Fellowship Program under Grant No. DGE1745016 and DGE2140739. Any opinions, findings, and conclusions or recommendations expressed in this material are those of the author(s) and do not necessarily reflect the views of the National Science Foundation.

Viraj Mehta was supported in part by US Department of Energy grants under contract numbers DE-SC0021414 and DE-AC02-09CH1146.

Willie Neiswanger was supported in part by NSF (#1651565), AFOSR (FA95501910024), ARO (W911NF-21-1-0125), CZ Biohub, and Sloan Fellowship.

In addition, this project has received support from the European Research Council (ERC) under the European Union's Horizon 2020 research and innovation programme grant No. 815943.

REPRODUCIBILITY STATEMENT

The proof of Theorem 4.2 is provided in Appendix A.2 and the proof of Corollary 5.1 is given in Appendix A.3. The supplementary material includes the source code for the experiments. It also includes a requirements file and README with full instructions on how to run the RL and BO experiments. Although we are not allowed to provide the data used for running the $\beta$ Tracking and $\beta$ + Rotation experiments at this time, all other experiments can be run using the provided code. Lastly, experimental details about the implementation and the environments used can be found in Appendix B.1 and Appendix B.2, respectively.

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

# A APPENDIX

## A.1 AUXILIARY RESULTS

**Lemma A.1.** *Let $t \in [T]$. Then, for every $(s, a) \in \mathcal{S} \times \mathcal{A}$,*

1. *If $\overline{Q}_h^t(s, a) \geq T_h^* \overline{Q}_{h+1}^t(s, a)$ holds for all $h \in [H]$, then $\overline{Q}_h^t(s, a) \geq Q_h^*(s, a)$ is true for all $h \in [H]$.*

2. *If $\underline{Q}_h^t(s, a) \leq T_h^* \underline{Q}_{h+1}^t(s, a)$ holds for all $h \in [H]$, then $Q_h^*(s, a) \geq \underline{Q}_h^t(s, a)$ is true for all $h \in [H]$.*

*Proof.* In order to prove part 1., let $s \in \mathcal{S}$ and $a \in \mathcal{A}$ and assume $\overline{Q}_h^t(s, a) \geq T_h^* \overline{Q}_{h+1}^t(s, a)$ for all $h \in [H]$ and $t \in [T]$. We prove $\forall h \in [H], \overline{Q}_h^t(s, a) \geq Q_h^*(s, a)$ by induction on $h = H, H - 1, \ldots, 1$. For the initial case $h = H$, we have

$$\overline{Q}_H^t(s, a) \overset{\text{assumption}}{\geq} T_H^* \overline{Q}_{H+1}^t(s, a) \tag{23}$$

$$\overset{\text{Def. of } T_h^*}{=} r_H(s, a) + \mathbb{E}_{s' \sim \mathbb{P}_H(\cdot | s, a)} \left[ \max_{a' \in \mathcal{A}} \overline{Q}_{H+1}^t(s', a') \right] \tag{24}$$

$$= r_H(s, a) \tag{25}$$

$$= Q_H^*(s, a). \tag{26}$$

For the inductive step, we assume that $Q_{h+1}^*(s, a) \leq \overline{Q}_{h+1}^t(s, a)$. Then,

$$Q_h^*(s, a) = T_h^* Q_{h+1}^*(s, a) \tag{27}$$

$$\overset{\text{Def. of } T_h^*}{=} r_h(s, a) + \mathbb{E}_{s' \sim \mathbb{P}_h(\cdot | s, a)} \left[ \max_{a' \in \mathcal{A}} Q_{h+1}^*(s', a') \right] \tag{28}$$

$$\overset{\text{inductive hypothesis}}{\leq} r_h(s, a) + \mathbb{E}_{s' \sim \mathbb{P}_h(\cdot | s, a)} \left[ \max_{a' \in \mathcal{A}} \overline{Q}_{h+1}^t(s', a') \right] \tag{29}$$

$$\overset{\text{Def. of } T_h^*}{=} T_h^* \overline{Q}_{h+1}^t(s, a) \tag{30}$$

$$\overset{\text{assumption}}{\leq} \overline{Q}_h^t(s, a). \tag{31}$$

This shows $\overline{Q}_h^t(s, a) \geq Q_h^*(s, a)$ for all $h \in [H]$ and thus concludes the proof of the first claim. The second part can be shown analogously.

$\square$

The following is a standard result that can be found in multiple works.

**Lemma A.2.** *Consider a kernel $k : \mathcal{X} \times \mathcal{X} \to \mathbb{R}$ such that $k(x, x) \leq 1$ for every $x \in \mathcal{X}$. Then for all $h \in [H]$ and $\lambda \geq 1$ we have*

$$\sum_{t=1}^{T} \sigma_h^t(s_h^t, a_h^t) \leq \sqrt{3\Gamma_k(T, \lambda)T}. \tag{32}$$

*Proof.* We can for example invoke the result of Lemma 3 in Bogunovic & Krause (2021) that in our notation reads as:

$$\sum_{t=1}^{T} \sigma_h^t(s_h^t, a_h^t) \leq \sqrt{\lambda^{-1}(2\lambda + 1)\Gamma_k(T, \lambda)T}, \tag{33}$$

for $\lambda > 0$. Setting $\lambda \geq 1$, we obtain

$$\sum_{t=1}^{T} \sigma_h^t(s_h^t, a_h^t) \leq \sqrt{3\Gamma_k(T, \lambda)T}. \tag{34}$$

$\square$

## A.2 PROOF OF THEOREM 4.2

Let $\hat{\pi}_T$ be the best-policy estimate returned by the algorithm. Recall the definition of

$$\pi_T^{*\geq h} := \left(\pi_{T,h'}^{*\geq h}\right)_{h'=1}^{H} := \begin{cases} \hat{\pi}_{T,h'} & \text{for } h' = 1, \dots, h-1 \\ \pi_{h'}^* & \text{for } h' = h, \dots, H \end{cases} \tag{35}$$

as the policy that equals our best-policy estimate $\hat{\pi}_T$ until step $h-1$ and then equals the optimal policy $\pi^*$.

We start the proof with the following useful lemma.

**Lemma A.3.** *Let $\hat{\pi}_T$ be a best-policy estimate, let $s \in \mathcal{S}$ be an initial state, and let $h \in [H]$. Using the notation from Eq. (35), we obtain*

$$V_1^{\pi_T^{*\geq h}}(s) - V_1^{\pi_T^{*\geq h+1}}(s) = \mathbb{E}_{a_1,\dots,s_h \text{ following } \hat{\pi}_T}\left[Q_h^*\big(s_h, \pi_h^*(s_h)\big) - Q_h^*\big(s_h, \hat{\pi}_{T,h}(s_h)\big)\big| s_1 = s\right].$$

*Proof.* To formally prove the lemma, we first explicitly express $V_1^{\pi_T^{*\geq h}}(s)$ and $V_1^{\pi_T^{*\geq h+1}}(s)$ for an arbitrary initial state $s \in \mathcal{S}$ as

$$V_1^{\pi_T^{*\geq h}}(s) \tag{36}$$

$$= \mathbb{E}_{a_1,\dots,s_H \text{ following } \pi_T^{*\geq h}|s_1=s}\left[\sum_{h'=1}^{H} r_{h'}(s_{h'}, a_{h'})\right] \tag{37}$$

$$= \mathbb{E}_{a_1,\dots,s_h \text{ following } \hat{\pi}_T|s_1=s}\left[\mathbb{E}_{a_h,\dots,s_H \text{ following } \pi^*|s_h}\left[\sum_{h'=1}^{H} r_{h'}(s_{h'}, a_{h'})\right]\right] \tag{38}$$

$$= \mathbb{E}_{a_1,\dots,s_h \text{ following } \hat{\pi}_T|s_1=s}\left[\sum_{h'=1}^{h} r_{h'}(s_{h'}, a_{h'}) + \mathbb{E}_{a_h,\dots,s_H \text{ following } \pi^*|s_h}\left[\sum_{h'=h+1}^{H} r_{h'}(s_{h'}, a_{h'})\right]\right], \tag{39}$$

and

$$V_1^{\pi_T^{*\geq h+1}}(s_1) \tag{40}$$

$$= \mathbb{E}_{a_1,\dots,s_H \text{ following } \pi_T^{*\geq h+1}|s_1=s}\left[\sum_{h'=1}^{H} r_{h'}(s_{h'}, a_{h'})\right] \tag{41}$$

$$= \mathbb{E}_{a_1,\dots,s_h \text{ following } \hat{\pi}_T|s_1=s}\left[\mathbb{E}_{a_h,s_{h+1} \text{ following } \hat{\pi}_T|s_h}\left[\mathbb{E}_{a_{h+1},\dots,s_H \text{ following } \pi^*|s_{h+1}}\right.\right.$$

$$\left.\left.\left[\sum_{h'=1}^{h} r_{h'}(s_{h'}, a_{h'}) + \sum_{h'=h+1}^{H} r_{h'}(s_{h'}, a_{h'})\right]\right]\right] \tag{42}$$

$$= \mathbb{E}_{a_1,\dots,s_h \text{ following } \hat{\pi}_T|s_1=s}\left[\sum_{h'=1}^{h} r_{h'}(s_{h'}, a_{h'})\right.$$

$$\left. + \mathbb{E}_{a_h,s_{h+1} \text{ following } \hat{\pi}_T|s_h}\left[\mathbb{E}_{a_{h+1},\dots,s_H \text{ following } \pi^*|s_{h+1}}\left[\sum_{h'=h+1}^{H} r_{h'}(s_{h'}, a_{h'})\right]\right]\right]. \tag{43}$$

Eqs. (37) and (41) use the definition of $V_1^\pi$, Eqs. (38) and (42) use the definition of $\pi_T^{*\geq h}$ and $\pi_T^{*\geq h+1}$ from Eq. (35), and Eqs. (39) and (43) use the property that integration is a linear operator.

Lemma A.3 then follows from Eqs. (39) and (43) as well as the definition of $Q_h^*$:

$$V_1^{\pi_T^{*\geq h}}(s_1) - V_1^{\pi_T^{*\geq h+1}}(s_1) = \mathbb{E}_{a_1,\dots,s_h \text{ following } \pi^T|s_1=s}\left[\sum_{h'=1}^{h} r_{h'}(s_{h'}, a_{h'}) - \sum_{h'=1}^{h} r_{h'}(s_{h'}, a_{h'})\right.$$

$$+ \mathbb{E}_{a_h,\ldots,s_H \text{ following } \pi^*|s_h} \left[ \sum_{h'=h+1}^{H} r_{h'}(s_{h'}, a_{h'}) \right]$$

$$- \mathbb{E}_{a_h, s_{h+1} \text{ following } \hat{\pi}_T|s_h} \left[ \mathbb{E}_{a_{h+1},\ldots,s_H \text{ following } \pi^*|s_{h+1}} \left[ \sum_{h'=h+1}^{H} r_{h'}(s_{h'}, a_{h'}) \right] \right] \right] \right] \tag{44}$$

$$= \mathbb{E}_{a_1,\ldots,s_h \text{ following } \hat{\pi}_T|s_1=s} \left[ Q_h^*(s_h, \pi_h^*(s_h)) - Q_h^*(s_h, \hat{\pi}_{T,h}(s_h)) \right]. \tag{45}$$

$\square$

We proceed with the proof by using the notation from Eq. (35). We can decompose the instantaneous regret for an arbitrary initial state $s \in \mathcal{S}$ as follows:

$$V_1^*(s) - V_1^{\hat{\pi}_T}(s) = V_1^{\pi_T^{*\geq 1}}(s) - V_1^{\pi_T^{*\geq H+1}}(s) \tag{46}$$

$$= \sum_{h=1}^{H} \left( V_1^{\pi_T^{*\geq h}}(s) - V_1^{\pi_T^{*\geq h+1}}(s) \right) \tag{47}$$

$$\overset{\text{Lemma A.3}}{=} \sum_{h=1}^{H} \mathbb{E}_{s_1,a_1,\ldots,s_h \text{ following } \hat{\pi}_T} \left[ Q_h^*(s_h, \pi_h^*(s_h)) - Q_h^*(s_h, \hat{\pi}_{T,h}(s_h)) \big| s_1 = s \right]. \tag{48}$$

The intuition behind Lemma A.3 used in Eq. (48) is as follows. Both $V_1^{\pi_T^{*\geq h}}(s)$ and $V_1^{\pi_T^{*\geq h+1}}(s)$ refer to the same random trajectory segment $(s_1, a_1, \ldots, s_h)$ until step $h$ (i.e., the same initial state and policy are used), which is captured as $\mathbb{E}_{s_1,a_1,\ldots,s_h \text{ following } \hat{\pi}_T}[\cdot]$. For the remaining steps $h, \ldots, H$, the policies only differ at step $h$, a property which is captured in the difference $Q_h^*(s_h, \pi_h^*(s_h)) - Q_h^*(s_h, \hat{\pi}_{T,h}(s_h))$.

Conditioning on the event in Lemma 4.1 holding true and by invoking Lemma A.1, we have that:

$$\underline{Q}_h^t(s, a) \leq Q_h^*(s, a) \leq \overline{Q}_h^t(s, a), \tag{49}$$

holds for every $h \in [H]$, $t \in [T]$, and $(s, a) \in \mathcal{S} \times \mathcal{A}$. Next, we proceed to bound $Q_h^*(\cdot, \pi_h^*(\cdot)) - Q_h^*(\cdot, \hat{\pi}_{T,h}(\cdot))$ from Eq. (48) uniformly on $\mathcal{S}$. We have:

$$Q_h^*(s, \pi_h^*(s)) - Q_h^*(s, \hat{\pi}_{T,h}(s)) \overset{\text{Eq. (49)}}{\leq} Q_h^*(s, \pi_h^*(s)) - \max_{t\in[T]} \underline{Q}_h^t(s, \hat{\pi}_{T,h}(s)) \tag{50}$$

$$\overset{\text{Def. of } \hat{\pi}_{T,h}}{=} Q_h^*(s, \pi_h^*(s)) - \max_{a\in\mathcal{A}} \max_{t\in[T]} \underline{Q}_h^t(s, a) \tag{51}$$

$$= \min_{t\in[T]} \left( Q_h^*(s, \pi_h^*(s)) - \max_{a\in\mathcal{A}} \underline{Q}_h^t(s, a) \right) \tag{52}$$

$$\overset{\text{Def. of } \pi_h^*}{=} \min_{t\in[T]} \left( \max_{a\in\mathcal{A}} Q_h^*(s, a) - \max_{a\in\mathcal{A}} \underline{Q}_h^t(s, a) \right) \tag{53}$$

$$\overset{\text{Eq. (49)}}{\leq} \min_{t\in[T]} \left( \max_{a\in\mathcal{A}} \overline{Q}_h^t(s, a) - \max_{a\in\mathcal{A}} \underline{Q}_h^t(s, a) \right) \tag{54}$$

$$\overset{\text{Eq. (11)}}{\leq} \min_{t\in[T]} \left( \max_{a\in\mathcal{A}} \overline{Q}_h^t(s_h^t, a) - \max_{a\in\mathcal{A}} \underline{Q}_h^t(s_h^t, a) \right) \tag{55}$$

$$\overset{\text{Eq. (12)}}{\leq} \min_{t\in[T]} \left( \overline{Q}_h^t(s_h^t, a_h^t) - \underline{Q}_h^t(s_h^t, a_h^t) \right) \tag{56}$$

$$\leq \frac{1}{T} \sum_{t=1}^{T} \left( \overline{Q}_h^t(s_h^t, a_h^t) - \underline{Q}_h^t(s_h^t, a_h^t) \right). \tag{57}$$

Next, for convenience we introduce the notation

$$d_h^t := \left( \overline{Q}_h^t(s_h^t, a_h^t) - T_h^* \overline{Q}_{h+1}^t(s_h^t, a_h^t) \right) + \left( T_h^* \underline{Q}_{h+1}^t(s_h^t, a_h^t) - \underline{Q}_h^t(s_h^t, a_h^t) \right), \qquad (58)$$

and obtain the following upper bound on $\overline{Q}_h^t(s_h^t, a_h^t) - \underline{Q}_h^t(s_h^t, a_h^t)$ (from Eq. (57)) for every $h \in [H], t \in [T]$:

$$\overline{Q}_h^t(s_h^t, a_h^t) - \underline{Q}_h^t(s_h^t, a_h^t) \overset{\text{Eq. (58)}}{=} d_h^t + T_h^* \overline{Q}_{h+1}^t(s_h^t, a_h^t) - T_h^* \underline{Q}_{h+1}^t(s_h^t, a_h^t) \qquad (59)$$

$$\overset{\text{Def. of } T_h^*}{=} d_h^t + \mathbb{E}_{s' \sim \mathbb{P}_h(\cdot | s_h^t, a_h^t)} \left( \max_{\overline{a} \in \mathcal{A}} \overline{Q}_{h+1}^t(s', \overline{a}) - \max_{\underline{a} \in \mathcal{A}} \underline{Q}_{h+1}^t(s', \underline{a}) \right) \qquad (60)$$

$$\leq d_h^t + \max_{s' \in \mathcal{S}} \left( \max_{\overline{a} \in \mathcal{A}} \overline{Q}_{h+1}^t(s', \overline{a}) - \max_{\underline{a} \in \mathcal{A}} \underline{Q}_{h+1}^t(s', \underline{a}) \right) \qquad (61)$$

$$\overset{\text{Eq. (11)}}{=} d_h^t + \left( \max_{\overline{a} \in \mathcal{A}} \overline{Q}_{h+1}^t(s_{h+1}^t, \overline{a}) - \max_{\underline{a} \in \mathcal{A}} \underline{Q}_{h+1}^t(s_{h+1}^t, \underline{a}) \right) \qquad (62)$$

$$\overset{\text{Eq. (12)}}{\leq} d_h^t + \left( \overline{Q}_{h+1}^t(s_{h+1}^t, a_{h+1}^t) - \underline{Q}_{h+1}^t(s_{h+1}^t, a_{h+1}^t) \right). \qquad (63)$$

Using the definition of $\underline{Q}_{H+1}^t$ and $\overline{Q}_{H+1}^t$ as the zero functions, we can unroll the recursive inequality from Eq. (63) and upper bound $\overline{Q}_h^t(s_h^t, a_h^t) - \underline{Q}_h^t(s_h^t, a_h^t)$ for every $h \in [H], t \in [T]$ as follows:

$$\overline{Q}_h^t(s_h^t, a_h^t) - \underline{Q}_h^t(s_h^t, a_h^t) \leq \sum_{h'=h}^{H} d_{h'}^t. \qquad (64)$$

$$\overline{Q}_h^t(s_h^t, a_h^t) - \underline{Q}_h^t(s_h^t, a_h^t) \qquad (65)$$

$$\leq \sum_{h'=h}^{H} \left[ \left( \overline{Q}_{h'}^t(s_{h'}^t, a_{h'}^t) - T_{h'}^* \overline{Q}_{h'+1}^t(s_{h'}^t, a_{h'}^t) \right) + \left( T_{h'}^* \underline{Q}_{h'+1}^t(s_{h'}^t, a_{h'}^t) - \underline{Q}_{h'}^t(s_{h'}^t, a_{h'}^t) \right) \right] \qquad (66)$$

$$\overset{\text{Lemma 4.1}}{\leq} \sum_{h'=h}^{H} 4\beta \sigma_{h'}^t(s_{h'}^t, a_{h'}^t). \qquad (67)$$

By substituting the bound from Eq. (67) in Eq. (57), and then in Eq. (48), we arrive at:

$$V_1^*(s) - V_1^{\hat{\pi}_T}(s) \leq 4\beta \sum_{h=1}^{H} \sum_{h'=h}^{H} \frac{1}{T} \sum_{t=1}^{T} \sigma_{h'}^t(s_{h'}^t, a_{h'}^t) \leq 2\sqrt{3}\beta H(H+1) \sqrt{\frac{\Gamma_k(T,\lambda)}{T}}, \qquad (68)$$

where the last inequality follows from Lemma A.2. Since Eq. (68) holds for any $s \in S$, we arrive at our main result:

$$\| V_1^* - V_1^{\hat{\pi}_T} \|_{\ell^\infty(\mathcal{S})} \leq 2\sqrt{3}\beta H(H+1) \sqrt{\frac{\Gamma_k(T,\lambda)}{T}}. \qquad (69)$$

### A.3 OFFLINE CONTEXTUAL BAYESIAN OPTIMIZATION

---
**Algorithm 2** AE-LSVI for offline contextual Bayesian optimization
---
**Require:** kernel function $k(\cdot, \cdot)$, exploration parameter $\beta > 0$, regularization parameter $\lambda$,
1: **for** $t = 1, \ldots, T$ **do**
2:      Obtain $\overline{Q}^t$ and $\underline{Q}^t$ from Equation (20)
3:      Choose $s^t \in \arg\max_{s \in S} \left[ \max_{a \in A} \overline{Q}^t(s, a) - \max_{a \in A} \underline{Q}^t(s, a) \right]$
4:      Choose $a^t \in \arg\max_{a \in \mathcal{A}} \overline{Q}^t(s^t, a)$
5:      Observe the reward $y_t = Q^*(s^t, a^t) + \eta_t$
6: **end for**
7: Output the policy estimate $\hat{\pi}_T$ such that $\hat{\pi}_T(\cdot) = \arg\max_{a \in \mathcal{A}} \max_{t \in [T]} \underline{Q}^t(s, a)$
---

Our algorithm for the offline contextual Bayesian optimization is presented in Algorithm 2. As a side observation, we note that similarly to Char et al. (2019), we can also simply incorporate context weights (i.e., given $\omega(s)$ that represents some weighting of context $s$ that may depend on the probability of seeing $s$ at evaluation time or the importance of $s$), in case they are available, into the proposed acquisition function, i.e.,

$$s^t \in \arg\max_{s \in S} \left[ \left( \max_{a \in A} \overline{Q}^t(s, a) - \max_{a \in A} \underline{Q}^t(s, a) \right) w(s) \right]. \tag{70}$$

### A.3.1 PROOF OF COROLLARY 5.1

*Proof.* In this proof, we condition on the event in Eq. (21) holding true. Similar arguments to the ones in Eq. (59) – Eq. (63) lead to the following for every $s \in \mathcal{S}$:

$$\max_{a \in \mathcal{A}} Q^*(s, a) - Q^*(s, \hat{\pi}_T(s)) \overset{\text{Eq. (21)}}{\leq} \max_{a \in \mathcal{A}} Q^*(s, a) - \max_{t \in [T]} \underline{Q}^t(s, \hat{\pi}_T(s)) \tag{71}$$

$$\overset{\text{Def. of } \hat{\pi}_T}{=} \max_{a \in \mathcal{A}} Q^*(s, a) - \max_{a \in \mathcal{A}} \max_{t \in [T]} \underline{Q}^t(s, a) \tag{72}$$

$$= \min_{t \in [T]} \left( \max_{a \in \mathcal{A}} Q^*(s, a) - \max_{a \in \mathcal{A}} \underline{Q}^t(s, a) \right) \tag{73}$$

$$\overset{\text{Eq. (21)}}{\leq} \min_{t \in [T]} \left( \max_{a \in \mathcal{A}} \overline{Q}^t(s, a) - \max_{a \in \mathcal{A}} \underline{Q}^t(s, a) \right) \tag{74}$$

$$\overset{\text{Def. of } s^t}{\leq} \min_{t \in [T]} \left( \max_{a \in \mathcal{A}} \overline{Q}^t(s^t, a) - \max_{a \in \mathcal{A}} \underline{Q}^t(s^t, a) \right) \tag{75}$$

$$\overset{\text{Def. of } a^t}{\leq} \min_{t \in [T]} \left( \overline{Q}^t(s^t, a^t) - \underline{Q}^t(s^t, a^t) \right) \tag{76}$$

$$\leq \frac{1}{T} \sum_{t=1}^{T} \left( \overline{Q}^t(s^t, a^t) - \underline{Q}^t(s^t, a^t) \right) \tag{77}$$

$$\overset{\text{Eq. (20)}}{=} \frac{1}{T} \sum_{t=1}^{T} \left( 2\beta_t \sigma^t(s^t, a^t) \right) \tag{78}$$

$$\leq \frac{2\beta_T}{T} \sum_{t=1}^{T} \sigma^t(s^t, a^t) \tag{79}$$

$$\overset{\text{Lemma A.2}}{\leq} \frac{2\beta_T \sqrt{3\Gamma_k(T, \lambda)}}{\sqrt{T}}. \tag{80}$$

Finally, by setting $\epsilon \geq \frac{2\beta_T \sqrt{3\Gamma_k(T, \lambda)}}{\sqrt{T}}$ and expressing it in terms of $T$, we arrive at the main result. $\square$

## B ADDITIONAL EXPERIMENTAL DETAILS

### B.1 IMPLEMENTATION

We use an exact Gaussian Process with a squared exponential kernel with learned scale parameters in each dimension for the value function regression in Eq. (4). We fit the kernel hyperparameters at each iteration using 1000 iterations of Adam (Kingma & Ba, 2014), maximizing the marginal log likelihood of the training data. We used the TinyGP package (Foreman-Mackey, 2021) built on top of JAX (Bradbury et al., 2018) in order to take advantage of JIT compilation. All experiments are conducted with a fixed bonus $\beta = 0.5$. We have empirically evaluated various settings of $\beta$ in Appendix B.3. We uniformly sample 1,000 points from the state space and evaluate them to find an approximate maximizer to the objective in Eq. (11).

**DDQN and BDQN.** For both of these methods we use networks with two hidden layers, each with 256 units. For the bootstrapped DQN, we use a network with 10 different heads, each representing a different $Q$ function. For each step collected during exploration, a corresponding mask is generated and added to the replay buffer that signifies which heads will train on this sample. Each $Q$ function has a probability of $0.5$ of being trained on each transition.

### B.2 ENVIRONMENTS

Each environment is defined with a native reward function taken from the literature. We established upper and lower bounds on the reward function value and used them to scale the reward function values to $[0, 1]$ so that our environments would match the theoretical results in this paper.

**Cartpole** We use a modified version of the cartpole environment from Mehta et al. (2022a) that has dense rewards as implemented in Wang et al. (2019). The state space is $4D$ and consists of the horizontal position and velocity of the cart as well as the angular position and velocity of the pole. $p_0$ in this environment is a normal distribution centered with the cart below the goal horizontally with the pole hanging down with very small variance. $p_0'$ is the same distribution displaced 5 meters to the right.

**Navigation** This is a $2D$ navigation problem with dynamics of the form $s_{t+1} = s_t + B(s_t)a_t$, where $B(t) = \begin{bmatrix} \sin(x_2/10) + 4 & 0 \\ 0 & 1.5\cos(x_1/10) - 2 \end{bmatrix}$. The goal is fixed at $\begin{bmatrix} 6 \\ 9 \end{bmatrix}$. We define $p_0$ to be the uniform distribution over the axis-aligned rectangle given by corners $\begin{bmatrix} -8 \\ -9 \end{bmatrix}$ and $\begin{bmatrix} -6 \\ -6 \end{bmatrix}$. We define $p_0'$ to be the uniform distribution over the axis-aligned rectangle given by corners $\begin{bmatrix} 1 \\ 4 \end{bmatrix}$ and $\begin{bmatrix} 3 \\ 7 \end{bmatrix}$. The reward function at every timestep is simply the negative $\ell_1$-distance between the agent and the goal.

**$\beta$ Tracking and $\beta$ + Rotation** Our two simulated plasma control problems are taken from Mehta et al. (2022a), which gives a thorough description of their relevance to the problem of nuclear fusion. At a high level, $\beta_N$ is a normalized plasma pressure ratio that is correlated with the economic output of a fusion reactor. Our $\beta$ **Tracking** environment aims to adjust the injected power in the reactions in order to achieve a target value of $\beta_N = 2\%$. The initial state distribution $p_0$ is taken from a set of real datapoints from shots on the DIII-D tokamak in San Diego. Our alternate initial state distribution $p_0'$ consists of simply adding $0.4$ to each component of a vector sampled from $p_0$. The reward function is the negative $\ell_1$-distance between the $\beta_N$ value and $2\%$. The dynamics are given by a learned model of the plasma state as introduced in Char et al. (2022).

The $\beta$ + **Rotation** environment is a more complex plasma control problem, introducing an additional actuator (injected torque) and an additional control objective (controlling plasma rotation). Control of plasma rotation is key to plasma stability and this is a reduced version of the realistic problem. This environment also uses a model from Char et al. (2022) for the dynamics, real plasma states for the initial state distribution $p_0$, and a fixed translation for the alternate initial state distribution $p_0'$. We also include a randomly drawn target for $\beta_N$ and rotation in the state space for every episode.

### B.3 EXPLORING $\beta$ VALUES

In the main paper, we report experiments with the exploration parameter $\beta = 0.5$ for all $t$. In this work, we do not explore principled methods of choosing $\beta$ and welcome future work in the area. In lieu of this, we provide an empirical analysis of the sensitivity of AE-LSVI to varying settings of $\beta$. We ran the AE-LSVI method on our evaluation environments as in the experiments in Table 2, where we allowed each method to collect 1,000 timesteps of data and evaluated the identified policies on the environments starting from a evaluation initial distribution $p_0'$ distinct from the initial distribution $p_0$. In Table 3 we observe that lower values of $\beta$ perform better because the confidence bounds seem too wide at higher settings, where the performance becomes similar to that

| Environment | $\beta = 0.25$ | $\beta = 0.5$ | $\beta = 1$ | $\beta = 2$ |
|---|---|---|---|---|
| Cartpole | $\mathbf{17.2 \pm 0.3}$ | $16.8 \pm 0.4$ | $16.3 \pm 0.5$ | $15.1 \pm 0.4$ |
| Navigation | $\mathbf{22.3 \pm 0.5}$ | $\mathbf{22.3 \pm 0.4}$ | $\mathbf{22.2 \pm 1.4}$ | $20.6 \pm 1.3$ |
| $\beta$ Tracking | $\mathbf{13.9 \pm 0.3}$ | $\mathbf{14.0 \pm 0.4}$ | $13.2 \pm 1.3$ | $13.4 \pm 0.7$ |
| $\beta$ + Rotation | $\mathbf{14.8 \pm 0.3}$ | $14.3 \pm 0.2$ | $14.1 \pm 0.9$ | $13.3 \pm 0.9$ |

Table 3: Average Return $\pm$ standard error of executing the identified best policy on the MDP starting from $p_0'$ over 5 seeds after collecting 1000 timesteps of data using the AE-LSVI method with varying values of the exploration parameter $\beta$.

of uncertainty sampling. Therefore, we recommend initially trying $\beta$-values around $0.2 - 0.5$ when applying AE-LSVI .

