# OpenReview forum: "Near-optimal Policy Identification in Active Reinforcement Learning"
_ICLR.cc/2023/Conference — ICLR 2023 notable top 5%_

### Official Review · Reviewer_Zsrd · 2022-10-25

**Confidence:** 3
**Correctness:** 3
**Technical Novelty And Significance:** 3
**Empirical Novelty And Significance:** 3
**Recommendation:** 8

**Clarity, Quality, Novelty And Reproducibility:**

The paper is clear and is of good quality. The proposed algorithms and the associated theoretical analysis are novel, to our knowledge. The implementation details are provided in the Appendix. Code is also provided in the supplementary materials.

**Strength And Weaknesses:**

The contributions and strengths of paper are listed above. Below, I list some points to discuss:

1. Page 3. It was mentioned that the goal in this setting is to identify an $\epsilon$-optimal policy while minimizing the number of necessary episodes $T$. This differs from the standard setting on cumulative regret minimization. The paper would benefit from adding more motivations to justify such an objective. Is this objective commonly used in the generative model setting?

2. Under the generative model setting, the agent has access to a simulator that can be used for adaptive data generation. The optimal policy that maximizes the cumulative reward in simulators is not necessarily maximizing that in real applications, due to potential difference from simulator and real-world environments. Would it be better to consider the more realistic setting with sim-to-real gap?

3. Instead of obtaining $s_h^t$ according to Equation (12), we can alternatively define some uncertainty measure for the value and directly selects $s_h^t$ that minimizes the uncertainty. Would this procedure work?

4. While kernel methods enjoy nice theoretical properties, their empirical implementation requires tuning of the kernel bandwidth. What kernel functions did you choose in the numerical experiments? Are the results sensitive to the choice of the bandwidth? How would you suggest practioners to select this hyper-parameter?

5. In continuous action setting, the author(s) proposed to first discretize the action space and then apply the proposed method. Can your methodology be extended to the continuous action setting? Alternatively, can we adaptively discretize the action space to improve the performance (see e.g., http://proceedings.mlr.press/v80/lee18b/lee18b.pdf; https://arxiv.org/pdf/2007.00717.pdf; https://openreview.net/pdf?id=rvKD3iqtBdk)?

6. The current methodology and theory are developed under the episodic MDP setting with time-varying dynamics. Can these results be extended to time-homogeneous MDP models?

**Summary Of The Paper:**

The paper is concerned with near-optimal policy identification in the generative model setting. The contributions of this paper includes:

(1) The development of the AE-LSVI algorithm for $\epsilon$-optimal identification;
(2) A concrete proposal of the algorithm based on the RKHS model;
(3) Establishment of the theoretical findings to justify the proposed algorithm;
(4) Empirical verification of the proposed algorithm.

**Summary Of The Review:**

The paper is well-written and studies an interesting research question. The proposed algorithm and the theoretical analysis are novel. I have some comments regarding the practical usage of the proposed method and the model setup. I hope the author(s) can address my concerns.

---

> ### Author Response · Authors · 2022-11-12
> **Authors' response to Reviewer Zsrd (1/2)**
>
> We thank the reviewer for the insightful review and thoughtful suggestions.  We have updated the manuscript in red color and we respond below to individual points from your review.
>
> > Q1: Page 3. It was mentioned that the goal in this setting is to identify an $\epsilon$-optimal policy while minimizing the number of necessary episodes. This differs from the standard setting on cumulative regret minimization. The paper would benefit from adding more motivations to justify such an objective. Is this objective commonly used in the generative model setting?
>
> We believe that  $\epsilon$-optimal policy identification is commonly used in the generative model setting [1, Chapter 3], whereas cumulative regret minimization is more often used in the online setting. The reason is that in the generative model setting the learner interacts with the environment by querying potentially arbitrary states (not necessarily complete episodes). Hence, it is less clear how to define the regret of the exploration strategy, compared to the optimal policy that completes a full episode. Moreover, when interacting with a simulator the typial goal is to identify a good policy rather than maximizing the reward during the exploration period.
>
> > Q2: Under the generative model setting, the agent has access to a simulator that can be used for adaptive data generation. The optimal policy that maximizes the cumulative reward in simulators is not necessarily maximizing that in real applications, due to potential difference from simulator and real-world environments. Would it be better to consider the more realistic setting with sim-to-real gap?
>
> We agree that the sim-to-real gap is another interesting challenge. In this work, we focus on understanding the sample complexity in the (realizable) generative model setting. We do feel that misspecified settings and bounding sim-to-real gap are both interesting and challenging directions for future work.
>
> >Q3: Instead of obtaining $s^t\_h$ according to Equation (12) [now Equation (11)], we can alternatively define some uncertainty measure for the value and directly selects $s^t\_h$ that minimizes the uncertainty. Would this procedure work?
>
>
>
> We believe that a strategy that queries the most uncertain state can be used. However, intuitively, it might unnecessarily focus on states that are uncertain, but for which we might already know the best action (and hence waste samples). On the other hand, our strategy is explicitly designed to focus on states for which the best action is the most uncertain (Eq. (11)). Finally, the strategy that explicitly minimizes the uncertainty would also suffer from the previous issue and, additionally, it would be non-myopic and hence more expensive to compute.
>
> > Q4: While kernel methods enjoy nice theoretical properties, their empirical implementation requires tuning of the kernel bandwidth. What kernel functions did you choose in the numerical experiments? Are the results sensitive to the choice of the bandwidth? How would you suggest practioners to select this hyper-parameter?
>
> As we discuss in Section B.1, we use a squared exponential kernel and fit the kernel hyperparameters via maximizing the marginal likelihood at every iteration of the least-squares procedure needed to estimate $\hat{Q}^t_h$. We added the useful detail that our kernel has learned scale parameters in each dimension. We found this a relatively easy way to avoid having to hand-tune the GP regression. We agree that this problem is difficult in general and difficulties in regression are possible.

---

> > ### Author Response · Authors · 2022-11-12
> > **Authors' response to Reviewer Zsrd (2/2)**
> >
> > > Q5: In continuous action setting, the author(s) proposed to first discretize the action space and then apply the proposed method. Can your methodology be extended to the continuous action setting? Alternatively, can we adaptively discretize the action space to improve the performance (see e.g., http://proceedings.mlr.press/v80/lee18b/lee18b.pdf; https://arxiv.org/pdf/2007.00717.pdf; https://openreview.net/pdf?id=rvKD3iqtBdk)?
> >
> > In principle, the AE-LSVI algorithm does not require a discrete action space. In our experiments, we discretized the action space only as an implementation detail in order to simplify the computation of the explored action $a^t_h := argmax\_{a \in \mathcal{A}} \max\_{t \in [t]} \underline{q}^t_h(s^t\_h, a^t\_h) $ (Eq. (12)). In addition, we wanted to use a simple discretization scheme so as to make for a clear comparison between methods. We believe that it may be possible to extend these ideas to an actor-critic method that handles continuous action spaces with a single policy evaluation, however, we leave this development to future work as it is likely a nontrivial effort.
> >
> > >Q6: The current methodology and theory are developed under the episodic MDP setting with time-varying dynamics. Can these results be extended to time-homogeneous MDP models?
> >
> > We think that extending the results to time-homogeneous MDP models is not trivial. For instance, it is less clear how to obtain upper and lower confidence bounds on the value function using dynamic programming (without essentially relaxing the analysis to the time-inhomogeneous result). We leave this as an important extension for future work.
> >
> > There has been recent prior work [3] which addresses regret minimization in the time-homogeneous setting using a construction based on the Eluder dimension of the function class. In this work, UCB-style bounds are presented which are appropriate for their setting. We believe that it would be possible to use these bonuses along with the ideas from our acquisition function in order to extend the method from [3] to a time-inhomogeneous MDP model.
> >
> >
> > [1] Alekh Agarwal, Nan Jiang, Sham M Kakade, & Wen Sun (2019). Reinforcement learning: Theory and algorithms.
> >
> > [2] Srinivas, N., Krause, A., Kakade, S.M., & Seeger, M.W. (2010). Gaussian Process Optimization in the Bandit Setting: No Regret and Experimental Design. ICML.
> >
> > [3] Wang, R., Salakhutdinov, R. R., & Yang, L. (2020). Reinforcement learning with general value function approximation: Provably efficient approach via bounded eluder dimension. NeurIPS.

---

> ### Author Response · Authors · 2022-11-23
> **Authors' response to Reviewer Zsrd – Follow-Up**
>
> We wanted to thank you again for the insightful suggestions, and kindly ask if you could please let us know if our answers are satisfactory or further elaboration is needed.

---

> > ### Comment · Reviewer_Zsrd · 2022-11-24
> > **Thank you**
> >
> > Thank you for the clarification! My comments have been addressed. I have increased my score based on the responses.

---

### Official Review · Reviewer_maWK · 2022-10-28

**Confidence:** 4
**Correctness:** 4
**Technical Novelty And Significance:** 3
**Empirical Novelty And Significance:** 3
**Recommendation:** 8

**Clarity, Quality, Novelty And Reproducibility:**

The paper is written in a very clear language. This is a very high quality work with prominent novelty. The reported results are reproducible.

**Strength And Weaknesses:**

Strengths:
   - Despite being a combination of multi-decade-old ideas, the eventual algorithm is simple, solid, and novel.
   - The theoretical results are thorough and sufficiently informative, while being a corollary of existing work.
   - The experiments are comprehensive enough and the reported results support the central claim of the paper.

Weaknesses:
   - The baselines chosen for comparison are rather arbitrary and not state-of-the-art.

**Summary Of The Paper:**

The paper introduces a policy search algorithm that combines kernelized LSTD with the "optimism in the face of uncertainty" approach, which is a well-known method in bandit algorithms. The paper analyzes the theoretical properties of the resulting algorithm and reports results on a number of simulated control environments.

**Summary Of The Review:**

Overall very solid work, which neatly combines the theory and practice of online policy search.

---

> ### Author Response · Authors · 2022-11-12
> **Authors' Response to Reviewer maWK**
>
> Thank you for your feedback. We are glad that you found the method “simple, solid, and novel”. We have added bootstrapped DQN as a more recent model-free RL baseline in order to allay your and other reviewers’ concerns. While we find that it is indeed more competitive than vanilla DDQN, it never outperforms AE-LSVI in Table 2.

---

> > ### Comment · Reviewer_maWK · 2022-11-28
> > **Keep my grade**
> >
> > Thanks for your response. Adding B-DQN has indeed been great.

---

### Official Review · Reviewer_v5Ka · 2022-11-04

**Confidence:** 3
**Correctness:** 3
**Technical Novelty And Significance:** 3
**Empirical Novelty And Significance:** 3
**Recommendation:** 8

**Clarity, Quality, Novelty And Reproducibility:**

The paper is clearly written and easy to follow. I would suggest to rewrite the title to contain "Best policy identification" instead of just "policy identification" as the former might be confused with "policy space identification" another branch of RL concerned with identifying the minimal space of policies containing the optimal (near-optimal) ones.

The related works is solid, it's almost exhaustive and clearly positions the paper in the literature. The only comments around it would be to include some previous work on best-arm identification (not only classical bandits) as it is fairly close to the setting considered in this paper (see also previous section of the review). Moreover I would suggest to the authors to consider moving this section after the preliminaries or even after the section 4, as the related works also compare AE-LSVI with other algorithms from the literature, and without reading section 2, 3 and 4 it might be hard for the reader to appreciate the differences.

The paper is fairly novel, it uses standard assumptions from the literature (RHKS) and similar ideas from the other research areas to define and analyze a novel algorithm.



**Strength And Weaknesses:**

Strengths:
 - The authors study the interesting problem of best policy identification in episodic reinforcement learning
 - The authors propose a fairly simple algorithm, easy to implement and with performance guarantees
 - The sample complexity bound does not scale with the state and action space sizes
 - The authors perform a thorough empirical evaluation explicitly identifying the limitations of the algorithms and the settings in which it performs better than the baselines

Weaknesses:
- The central point of the algorithm is choosing states to sample from he generative model based on the perfomance gap, i.e. the difference between the upper and lower bounds on the V-function of the state. The authors should  cite other works that performs selection on similar metrics based on this uncertainty. For example, there is a large body of work in the best-arm identification literature that uses the difference between upper and lower bounds to select which arm to choose, [1], [2] etc.
- The paper does use a fairly common assumption from the literature, but nonetheless it should be more self contained. I would have expected to see a discussion on how restrictive is the RHKS assumption as well as a comparison of the guarantees provided non only with methods taking the same assumption but also with other method that make less or more limiting assumptions. For example, a comparison with methods that assume linearity of the reward and value functions would improve the clarity of the paper.
- A bit more discussion on why the bound presented loses the dependency on the state and action space sizes might also improve the paper.  Especially when comparing with the algorithm presented in Cher et. al. 2019 at the end of Section 4 which also makes the RHKS assumption, has a dependency on $\Gamma_k$ but also on the state and action spaces.
- The algorithms adds an additional parameter $\beta_t$, fairly important since it is used to define the upper and lower bound on Q-values, central for the "exploration" method used in AE-LSVI. Unfortunately, no discussion on the selection of this hyperparameter is provided. The authors present all the results with the value 0.5 (why 0.5). Instead, an empirical evaluation of the effect off the value of $\beta_t$ is warranted.
- While I am mostly positive on the experimental section, I am surprised with the choice of DDQN as a baseline. DDQN was chosen as a SOTA method for online RL, but there are more advanced methods right now for online RL, especially it would have been beneficial to see other online RL methods that use uncertainty estimates just like AE-LSVI like [3] [4] or [5]. Any of them would make a better baseline then DDQN but especially [3] since it makes decisions also based on upper and lower bounds on Q.

[1] Victor Gabillon, Mohammad Ghavamzadeh, Alessandro Lazaric, Sébastien Bubeck, Multi-Bandit Best Arm Identification, NeurIPS 2022

[2] Marta Soare, Alessandro Lazaric, Rémi Munos, Best-Arm Identification in Linear Bandits, NeurIPS 2014

[3] Ted Moskovitz, Jack Parker-Holder, Aldo Pacchiano, Michael Arbel, Michael I. Jordan, Tactical Optimism and Pessimism for Deep Reinforcement Learning

[4] Ian Osband, Charles Blundell, Alexander Pritzel, Benjamin Van Roy, Deep Exploration via Bootstrapped DQN

[5] Alberto Maria Metelli, Amarildo Likmeta, Marcello Restelli, Propagating Uncertainty in Reinforcement Learning via Wasserstein Barycenters


**Summary Of The Paper:**

The paper presents a novel algorithm, based on LSVI to perform best-policy identification uniformly across the state space. The method is based on a RHKS assumption on the reward and value functions and performs targeted 'exploration' based on the performance gaps defined as the difference between upper and lower bounds of the action value function. The authors study theoretically the performance of the algorithm in function of $\Gamma_k$, a property of the specific kernels used and give specific bounds for some families of frequently used kernels. Moreover, the authors perform an empirical evaluation on several domains, comparing AE-LSVI with algorithms from SOTA, explicitly showing the conditions under which this method outperforms the baselines.

**Summary Of The Review:**

Overall I am positive about the paper. The issues I describe all all fairly minor and solvable. I would increase my score if my concerns are adequately addressed. (See previous sections for details)

---

> ### Author Response · Authors · 2022-11-12
> **Authors’ Response to Reviewer v5kA (2/2)**
>
> We hope that we have addressed all comments and concerns. If so, we kindly ask the reviewer to reconsider the provided score as mentioned in the review.
>
> [6] Pihe Hu, Yu Chen, & Longbo Huang (2022). Nearly minimax optimal reinforcement learning with linear
> function approximation. ICML.
>
> [7] Alekh Agarwal, Nan Jiang, Sham M Kakade, & Wen Sun (2019). Reinforcement learning: Theory and algorithms.

---

> ### Author Response · Authors · 2022-11-12
> **Authors’ Response to Reviewer v5kA (1/2)**
>
> We thank the reviewer for the positive feedback and suggestions for improvement. We have updated the manuscript in red color and we respond below to individual points from your review.
>
> > The central point of the algorithm is choosing states to sample from the generative model based on the performance gap, i.e. the difference between the upper and lower bounds on the V-function of the state. The authors should cite other works that performs selection on similar metrics based on this uncertainty. For example, there is a large body of work in the best-arm identification literature that uses the difference between upper and lower bounds to select which arm to choose, [1], [2] etc.
>
> Thank you for the suggestions. We believe that while the BAI works are certainly relevant, they provide instance-dependent performance guarantees (i.e. gap-dependent) while our bounds hold even in the worst-case (i.e. for any function within the considered function class). In the updated manuscript, we will cite those works and point out this crucial difference. We also note that our acquisition rule is different than the ones in the referenced works.
>
> > The paper does use a fairly common assumption from the literature, but nonetheless it should be more self-contained. I would have expected to see a discussion on how restrictive is the RHKS assumption as well as a comparison of the guarantees provided non only with methods taking the same assumption but also with other methods that make less or more limiting assumptions. For example, a comparison with methods that assume linearity of the reward and value functions would improve the clarity of the paper.
>
> We have further elaborated the RKHS assumptions (see the paragraph below “Assumption 1”) and compared our sample size guarantees with results built upon the _Linear MDP_ assumption [6] or the _Linear Bellman Completeness_ assumption [7, Theorem 3.3] (see the last paragraph of the “Theoretical Results” section). As a side note: Linear MDPs (i.e., both rewards functions and transition kernels are linear) satisfy the RKHS assumption (Assumption 1) when a linear kernel is chosen.
>
> > A bit more discussion on why the bound presented loses the dependency on the state and action space sizes might also improve the paper. Especially when comparing with the algorithm presented in Char et. al. 2019 at the end of Section 4 which also makes the RHKS assumption, has a dependency on $\Gamma_k$ but also on the state and action spaces.
>
> In the Bayesian optimization setting, our bound depends only poly-logarithmically on $|S||A|$ (since $\beta_T$ depends on it), which is an improvement over $O(|S|^3 |A|)$ in Char et. al. 2019. We have now made this more precise in the manuscript. We believe that such a difference is due to a more careful analysis performed in our paper. For example, the analysis in Char et. al. 2019 relies on adaptive submodularity techniques that are not used in our paper.
>
> > The algorithms add an additional parameter $\beta\_t$ , fairly important since it is used to define the upper and lower bound on Q-values, central for the "exploration" method used in AE-LSVI. Unfortunately, no discussion on the selection of this hyperparameter is provided. The authors present all the results with the value 0.5 (why 0.5). Instead, an empirical evaluation of the effect off the value of $\beta_t$  is warranted.
>
> We agree that the value of $\beta_t$ is important and we did not do an extensive hyperparameter search. In order to give more context for values of $\beta_t$, we added Section B.3, where we evaluate AE-LSVI with various settings of $\beta_t$. We find that lower values of $\beta_t$ seem to perform better on our benchmark tasks.
>
> > While I am mostly positive on the experimental section, I am surprised with the choice of DDQN as a baseline. DDQN was chosen as a SOTA method for online RL, but there are more advanced methods right now for online RL, especially it would have been beneficial to see other online RL methods that use uncertainty estimates just like AE-LSVI like [3] [4] or [5]. Any of them would make a better baseline then DDQN but especially [3] since it makes decisions also based on upper and lower bounds on Q.
>
> Thank you for your suggestions on additional baselines. We added bootstrapped DQN as a baseline and, while we do find that it is indeed more competitive than vanilla DDQN, it never outperforms AE-LSVI in Table 2.
>
> > Moreover I would suggest to the authors to consider moving [the related work] section after the preliminaries or even after the section 4, as the related works also compare AE-LSVI with other algorithms from the literature, and without reading section 2, 3 and 4 it might be hard for the reader to appreciate the differences.
>
> We have moved the related work section after the theoretical results.

---

> > ### Comment · Reviewer_v5Ka · 2022-11-14
> > **Response to authors**
> >
> > I thank the author for the implemented changes, especially the additional baseline and the empirical study of the impact of $\beta$ on the perfomance. I will increase my score to 8 as my concerns were addressed. I would additionally suggest the authors to explicitly reference the experiments on the value of $\beta$ in the main paper, as it would help readers interested in implementing the method.

---

> > > ### Author Response · Authors · 2022-11-15
> > > **Authors' response to Reviewer v5Ka**
> > >
> > > As suggested, we will reference our $\beta$-experiments in the main text. We thank the reviewer for the useful advice and score update!

---

### Decision · Program_Chairs · 2023-01-20

**Decision:**

Accept: notable-top-5%

**Justification For Why Not Higher Score:**

N/A

**Justification For Why Not Lower Score:**

While the paper could also be accepted as a spotlight, I would not go below that level given the relevance of the contributions.

**Metareview: Summary, Strengths And Weaknesses:**

The paper proposes a novel active reinforcement learning algorithm based on a kernelized version of LSVI, which provably finds a near-optimal policy uniformly over the entire state space.
After reading each other's reviews and the authors' feedback, the reviewers agree that the paper studies an interesting problem and proposes a sound solution, which is analyzed both theoretically and empirically. The main weak point raised by the reviewers is the lack of competitive baselines in the experiments, which has been partly addressed by the authors through the addition of Bootstrapped DWN.
Overall, the paper has solid contributions and deserves to be published.

**Note From Pc:**

if the above contains the word "oral" or "spotlight" please see: "oral" presentation means -> notable-top-5% and "spotlight" means -> notable-top-25%. As stated in our emails, we are disassociating presentation type from AC recommendations

**Summary Of Ac-Reviewer Meeting:**

N/A